# Mapping the serum proteome to neurological diseases using whole genome sequencing

Grace Png [1,2✉], Andrei Barysenka[1], Linda Repetto[3], Pau Navarro [4], Xia Shen [3,5,6], Maik Pietzner [7], Eleanor Wheeler [7], Nicholas J. Wareham [7], Claudia Langenberg [7,8], Emmanouil Tsafantakis[9], Maria Karaleftheri[10], George Dedoussis[11], Anders Mälarstig [12,13], James F. Wilson [3,4], Arthur Gilly[1] & Eleftheria Zeggini [1,2✉]

Despite the increasing global burden of neurological disorders, there is a lack of effective diagnostic and therapeutic biomarkers. Proteins are often dysregulated in disease and have a strong genetic component. Here, we carry out a protein quantitative trait locus analysis of 184 neurologically-relevant proteins, using whole genome sequencing data from two isolated population-based cohorts ($N = 2893$). In doing so, we elucidate the genetic landscape of the circulating proteome and its connection to neurological disorders. We detect 214 independently-associated variants for 107 proteins, the majority of which (76%) are cis-acting, including 114 variants that have not been previously identified. Using two-sample Mendelian randomisation, we identify causal associations between serum CD33 and Alzheimer's disease, GPNMB and Parkinson's disease, and MSR1 and schizophrenia, describing their clinical potential and highlighting drug repurposing opportunities.

[1] Institute of Translational Genomics, Helmholtz Zentrum München – German Research Center for Environmental Health, Neuherberg, Germany. [2] TUM School of Medicine, Technical University of Munich and Klinikum Rechts der Isar, Munich, Germany. [3] Centre for Global Health Research, Usher Institute, University of Edinburgh, Edinburgh, UK. [4] MRC Human Genetics Unit, Institute of Genetics and Molecular Medicine, University of Edinburgh, Edinburgh, UK. [5] Greater Bay Area Institute of Precision Medicine (Guangzhou), Fudan University, Guangzhou, China. [6] Department of Medical Epidemiology and Biostatistics, Karolinska Institute, Stockholm, Sweden. [7] MRC Epidemiology Unit, University of Cambridge, Cambridge, UK. [8] Computational Medicine, Berlin Institute of Health (BIH), Charité University Medicine, Berlin, Germany. [9] Anogia Medical Centre, Anogia, Greece. [10] Echinos Medical Centre, Echinos, Greece. [11] Department of Nutrition and Dietetics, School of Health Science and Education, Harokopio University of Athens, Athens, Greece. [12] Department of Medicine, Karolinska Institute, Solna, Sweden. [13] Emerging Science & Innovation, Pfizer Worldwide Research, Development and Medical, Cambridge, MA, USA. ✉email: grace.png@helmholtz-muenchen.de; eleftheria.zeggini@helmholtz-muenchen.de

Neurological disorders are the leading cause of disability worldwide, accounting for 276 million disability-adjusted life years (DALY) globally in 2016[1]. This burden is continuously increasing with growing and ageing populations[2], emphasising the need for better prevention and treatment strategies. Multiple genetics and genomics efforts have established that these diseases have a substantial genetic component[3,4]. Elucidating their genetic architecture can, therefore, help to forward our understanding of their aetiology by identifying causal disease mechanisms, thus opening a path towards clinical translation.

Due to their heterogeneity and overlapping clinical features, neuropsychiatric disorders such as schizophrenia and bipolar disorder are often misdiagnosed[5], while others with more distinct symptoms, such as Alzheimer's disease (AD), lack effective drugs and accessible biomarkers that can detect early disease[6]. The human serum proteome is an especially valuable resource of potential biomarkers for these highly polygenic disorders. As proteins are often dysregulated in disease, studying protein quantitative trait loci (pQTLs), which are genetic variants associated with protein expression levels, can help to bridge existing knowledge gaps. Most pharmaceutical drugs also target proteins, further increasing their actionability.

By implementing statistical methods that leverage relevant biomedical data, such as causal inference and colocalisation analysis, pQTLs can be used to determine causality and to identify disease pathways. For example, in a study focused on neurologically relevant proteins[7], a pQTL for serum PVR mapping to the *PVR* gene (*cis*-pQTL), was found to be causally associated with AD through Mendelian randomisation analysis. Through similar methods, a recent brain proteome-wide association (PWAS) and pQTL study[8] identified five genes causal for AD at high confidence, of which four were novel. By validating known AD loci and identifying new causal genes, these studies demonstrate proof-of-concept.

Here, we aimed to identify biomarkers of neurological traits and enhance insight into disease pathways, by carrying out a pQTL analysis of 184 neurologically relevant serum proteins. The main advantage of serum proteins is that they are easily accessible, both as drug targets and diagnostic biomarkers. We use whole-genome sequencing (WGS) to capture the entire allele frequency spectrum in 2,893 samples from two Greek population-based cohorts, MANOLIS and Pomak. Association analysis was first carried out individually for each cohort, followed by a meta-analysis. Specifically, proteins were quantified using Olink's proximity extension assay (PEA) and comprised established or potential markers of neurobiological processes. Using WGS, we were able to detect both rare and common pQTL variants. We then investigated the relevance of the discovered pQTLs to neurological diseases and highlight biomarkers of high diagnostic or prognostic potential, identify drug repositioning opportunities, and describe pathways relevant to neurological traits.

## Results

**Protein QTL discovery**. For the 184 neurologically relevant proteins analysed, we detect 214 independently-associated pQTLs ($P < 1.05 \times 10^{-10}$; 'Methods' section) for 107 proteins from the meta-analysis, following conditional testing (Fig. 1 and Supplementary Data 1). Loci were classified into *cis* and *trans*: *cis*-acting pQTLs, which are defined as variants residing within 1 Mb upstream or downstream of the protein-encoding gene, are likely to regulate protein expression directly at the transcriptional level, while *trans*-pQTLs are likely to act through intermediaries to modulate protein levels. We observe 162 (75.7%) *cis*-acting pQTLs for 91 proteins, and 52 (24.3%) *trans*-acting pQTLs for 38 proteins. A total of 22 proteins had both *cis* and *trans*-acting pQTLs (Fig. 2b).

Sixteen proteins have only *trans*-pQTLs, 13 of which have pQTLs only in pleiotropic loci. We find altogether 30 variants arising at known pleiotropic loci, including those near or within *KLKB1, ABO, F12, VTN*, and the HLA region on chromosome 6. These are loci that influence the levels of multiple proteins; the most pleiotropic being loci at *KLKB1* and *ABO*, affecting 11 and 12 proteins, respectively. These have been identified in published pQTL studies and are not restricted to neurologically relevant proteins[9–12]. *ABO* is the most extensively studied among these pleiotropic loci, and is known for its role in blood coagulation processes and determining the ABO blood types. In particular, we detect the missense variant rs8176747 affecting ADAM15, IL3RA, and KIRREL2 protein levels. rs8176747 is among the variants routinely used to determine blood group phenotype[13], which has been associated with multiple diseases, mainly of cardiovascular relevance. As proteins such as ABO are connected to large signalling networks, changes in their structure or expression levels could influence multiple downstream substrates, hence explaining their pleiotropy.

We identify 33 sequence variant-protein level independent associations for 15 proteins that have not been investigated for pQTLs before (Table 1). For the remaining 92 proteins, we identify 72 novel *cis*-pQTL variants, and 15 novel *trans*-pQTL variants, excluding those at known pleiotropic loci. We define novelty if no variants within 2 Mb have been previously reported in serum pQTL studies, or if associations remain significant after conditioning on established pQTLs.

Eight of the proteins we studied here have also been investigated in a pQTL study in cerebrospinal fluid (CSF)[14]. We replicate six of these *cis*-pQTLs in serum: for CD33, GPNMB, LEPR, NAAA, SIGLEC-9, and TDGF1. Additionally, we find novel *cis*-pQTLs for CD33 and GPNMB, and *trans*-pQTLs for NAAA and SIGLEC-9, which had not been detected in CSF. The observed replication of CSF pQTLs indicates that the expression of these proteins in serum and CSF are governed by a shared genetic mechanism.

Of the identified independent pQTLs, 185 (86%) are common-frequency variants (minor allele frequency [MAF] > 5%), 25 (12%) are low-frequency (MAF 1–5%) and four (2%) are rare (MAF < 1%) (Fig. 2a). Eight of the low-frequency or rare pQTLs (all *cis* signals) have not been reported before, despite the proteins having been analysed in past studies, demonstrating the advantage of using whole-genome sequencing-based analysis to capture the full MAF spectrum.

**Gene expression QTL colocalisation**. Colocalisation analysis is used to test if independent association signals from two traits share the same causal variant. When comparing protein with gene expression levels, positive colocalisation is indicative of a shared regulatory mechanism, thereby acting as orthogonal validation. Through testing for colocalisation of neurological pQTLs with gene expression QTLs (eQTLs) from multiple tissues (GTEx), our results also identify disease-relevant tissues where gene expression correlates with serum protein expression. For *cis*-acting pQTLs, analysis was carried out between protein expression and the expression of the encoding gene, in all available tissues. Sixty-four (69%) *cis*-pQTLs colocalised strongly (colocalisation posterior probability 4 [CLPP4] > 0.8; 'Methods' section) with gene expression in at least one tissue, with 11 (12%) in whole blood, and 21 (23%) in various parts of the brain (Supplementary Data 4). This indicates that for these loci, the causal variant influences both gene and protein expression, therefore supporting transcriptional regulation as the mechanism underpinning variation in protein expression levels.

For *trans*-pQTLs, positive colocalisation between a pQTL and an eQTL at a distal gene increases the likelihood that the two gene

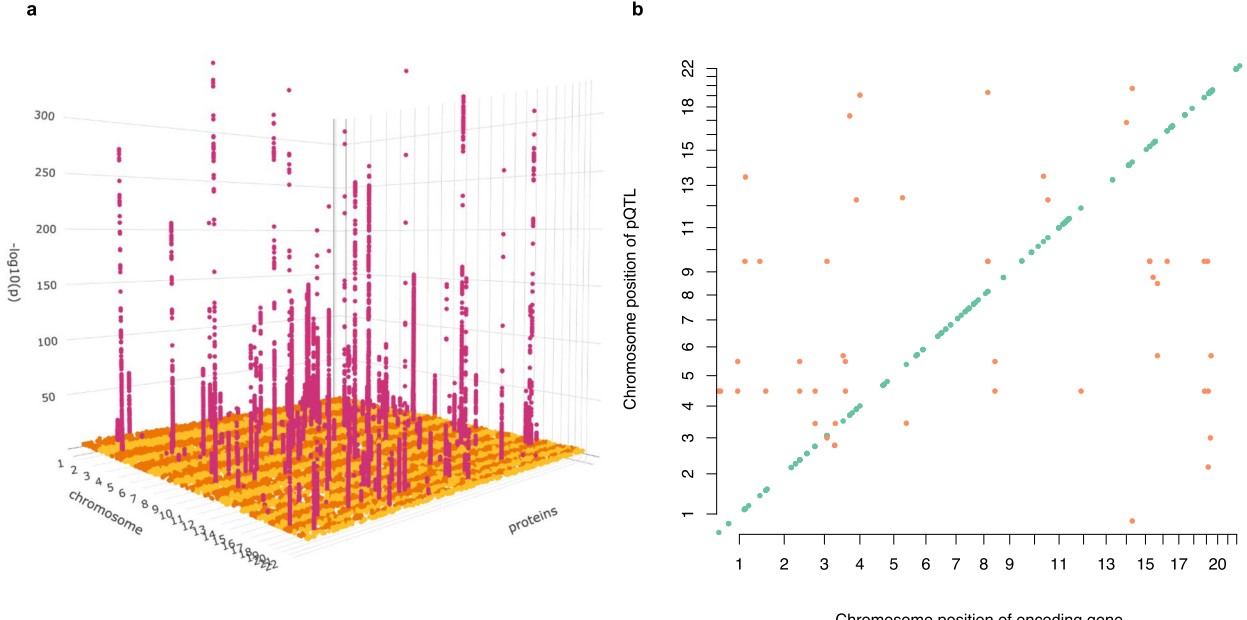

**Fig. 1 pQTL signals for 107 serum proteins from Olink neurology and neuro-exploratory panels. a** 3D Manhattan plot of detected pQTLs. The *x* axis represents each of the 107 proteins; the *y* axis represents the chromosome location of each signal; and the *z* axis represents the −log10 *p*-values of each association signal. **b** Scatterplot of pQTL variant location against the location of the gene encoding the target protein. Each dot represents an independent variant. *Cis*-pQTLs are coloured in teal, while *trans*-pQTLs are in orange.

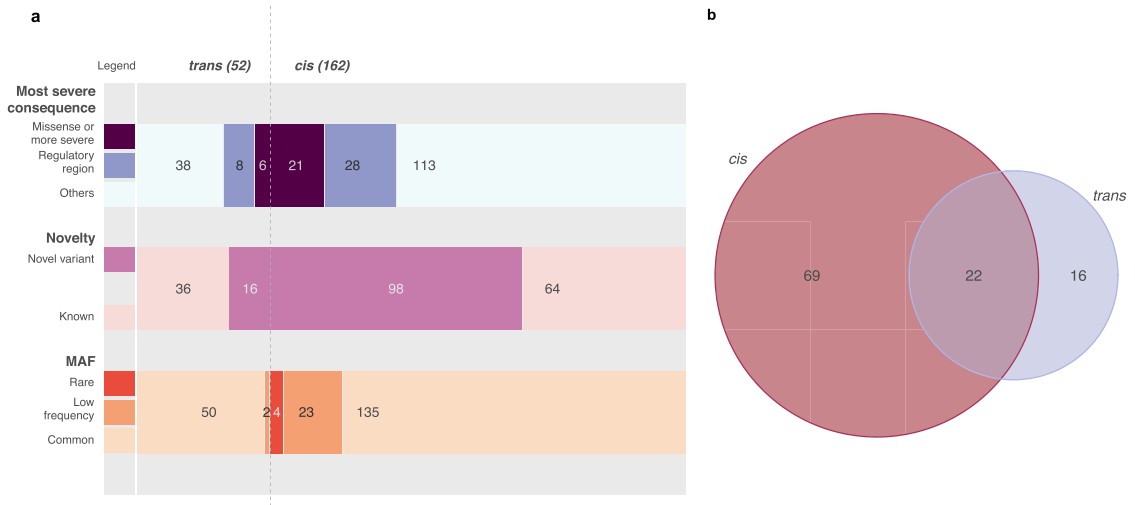

**Fig. 2 Overall genetic architecture of 107 serum proteins of neurological relevance. a** A total of 214 independent variants were detected. *Cis*-acting variants were defined as variants lying within 1 Mb upstream and downstream of the gene encoding the target protein, while *trans*-acting variants are variants that lie outside of this region. Most severe consequence was determined by Ensembl's variant effect predictor (VEP). Effects more than missense included 'stop_gained', 'frameshift_variant', and 'splice_acceptor_variant' in our dataset; 'Regulatory region' variants include '[3/5]_primeUTR_variant', 'TF_binding_site_variant', 'splice_region_variant', and 'regulatory_region_variant'; while 'Others' comprises mostly intergenic and intronic variants. Novelty was assessed by cross-referencing published summary statistics from other pQTL studies (Supplementary Data 2). Known pleiotropic loci were not considered novel. Rare, low-frequency and common variants were defined as variants with minor allele frequency (MAF) < 1%, MAF 1–5%, and MAF > 5%, respectively. **b** Number of proteins for which we detected only *cis*-pQTLs, *trans*-pQTLs, or both.

products map to the same regulatory pathway (Supplementary Note 1 and Supplementary Fig. 2). Colocalisation analysis was performed between protein traits and expression of genes within 2 Mb of the *trans*-acting variant. We detect 36 (75%) signals that colocalise with the expression of at least one gene in their vicinity, with three (6%) in whole blood and 30 (62%) in the brain (Supplementary Data 4). As proof-of-concept, we find known receptor-ligand pairs such as a *trans* signal for the KIR2DL3 (killer cell immunoglobulin-like receptor 2DL3) protein colocalising with

the expression of *HLA-C* in multiple tissues (22 tissues; CLPP4 > 0.78). KIR2DL3 is an inhibitory receptor for HLA-C, and is responsible for preventing natural killer cells from killing healthy cells[15].

The analysis also enabled the identification of new protein links. For example, we observe a *trans*-pQTL for SMPD1 (sphingomyelin phosphodiesterase; rs10745925; MAF = 0.333; $P = 7.75 \times 10^{-23}$; BETA = −0.2805; SE = 0.0285) that colocalises strongly with the expression of *GNPTAB* in the liver (CLPP4: 0.89), and moderately in

**Table 1 Independent pQTL variants for proteins that are being analysed for the first time.**

| Protein | Variant | MAF | BETA | S.E. | P-value | rsID |
|---|---|---|---|---|---|---|
| ADGRB3 | chr6:68956792 | 0.1576 | 0.89 | 0.032 | 8.44E−170 | rs1932618 |
| ADGRB3 | chr6:68962147 | 0.3461 | 0.4947 | 0.0262 | 2.31E−79 | rs3798971 |
| ADGRB3 | chr6:68968025 | 0.3468 | 0.8342 | 0.0225 | 2.83E−301 | rs1953613 |
| CD302 | chr2:159745359 | 0.1016 | −0.4303 | 0.0436 | 5.34E−23 | rs5002908 |
| CD302 | chr2:159773858 | 0.3098 | 0.3731 | 0.0281 | 3.64E−40 | rs1553790820 |
| CDH17 | chr9:133253728 | 0.0918 | −0.6534 | 0.0462 | 1.70E−45 | rs10793962 |
| CDH17 | chr9:133264504 | 0.3431 | −0.3879 | 0.028 | 1.19E−43 | novel |
| CDH17 | chr19:48703205 | 0.4516 | −0.386 | 0.0264 | 2.25E−48 | rs681343 |
| CDH17 | chr8:94194571 | 0.4782 | −0.2672 | 0.0276 | 3.61E−22 | rs56129387 |
| CDH17 | chr8:94130944 | 0.4847 | 0.2889 | 0.0267 | 3.21E−27 | rs1051624 |
| GGT5 | chr22:24232046 | 0.0064 | −2.3071 | 0.1696 | 3.75E−42 | rs200519116 |
| GGT5 | chr22:24235780 | 0.1923 | −0.3614 | 0.0326 | 1.52E−28 | rs6004108 |
| GGT5 | chr22:24247481 | 0.2015 | −0.3049 | 0.0317 | 7.33E−22 | rs5760275 |
| IFI30 | chr19:18172691 | 0.2613 | 0.3604 | 0.0295 | 2.10E−34 | rs273266 |
| IMPA1 | chr8:81652967 | 0.3331 | 0.3338 | 0.0278 | 3.41E−33 | rs2142316 |
| KIR2DL3 | chr19:54744273 | 0.0665 | 0.8024 | 0.0574 | 2.11E−44 | rs10414825 |
| KIR2DL3 | chr19:54743423 | 0.2167 | 0.6973 | 0.0299 | 5.70E−120 | rs11667532 |
| KIR2DL3 | chr6:31272403 | 0.266 | 0.5934 | 0.0307 | 1.71E−83 | rs2524093 |
| KLB | chr17:68883786 | 0.0268 | −0.556 | 0.0849 | 5.79E−11 | rs34931250 |
| KLB | chr4:39431127 | 0.3249 | −0.4173 | 0.0265 | 5.44E−56 | rs2926042 |
| KLB | chr4:39447786 | 0.333 | 0.7642 | 0.025 | 1.17E−205 | rs12513342 |
| LTBP3 | chr11:65572664 | 0.0527 | 0.5989 | 0.058 | 5.49E−25 | rs10896017 |
| LTBP3 | chr11:65575510 | 0.2504 | 0.253 | 0.0299 | 2.68E−17 | rs67924081 |
| NDRG1 | chr5:177412889 | 0.2384 | 0.2707 | 0.0318 | 1.67E−17 | rs2731674 |
| NDRG1 | chr4:186235350 | 0.4738 | 0.2847 | 0.0263 | 2.23E−27 | novel |
| PSG1 | chr19:42929524 | 0.02 | 0.7883 | 0.087 | 1.32E−19 | rs146569565 |
| PSG1 | chr19:42872373 | 0.1525 | −0.3243 | 0.033 | 7.79E−23 | rs60887906 |
| PSG1 | chr19:42881078 | 0.192 | 0.8012 | 0.0267 | 5.72E−198 | rs2005772 |
| RBKS | chr2:27858572 | 0.009 | 1.9199 | 0.1685 | 4.54E−30 | rs140948699 |
| SNCG | chr10:86945549 | 0.2564 | 0.934 | 0.0217 | 3.24E−403 | rs3750822 |
| TPPP3 | chr16:67267204 | 0.0813 | −0.3312 | 0.0483 | 6.86E−12 | rs7200971 |
| VSTM1 | chr19:54062922 | 0.1819 | −0.8967 | 0.0309 | 9.59E−185 | rs8111849 |
| VSTM1 | chr19:54042277 | 0.3968 | 0.8847 | 0.0218 | 4.71E−359 | rs2433724 |

other tissues (CLPP4 = 0.58 [oesophagus mucosa]; 0.57 [stomach]; 0.54 [adrenal gland]). SMPD1 is a lipid hydrolase involved in multiple cell processes; whereas *GNPTAB* encodes subunits of GlcNAc-1-phosphotransferase, which is involved in the synthesis of mannose-6-phosphate (M6P). SMPD1 exists in two forms: secreted and lysosomal. Its lysosomal form is transported via the M6P receptor pathway, therefore supporting the observed SMPD1-GNPTAB interaction. Moreover, we find that the minor allele is associated with a decrease in circulating SMPD1 and an increase in *GNPTAB* expression. This could be a result of increased M6P tagging, which targets a disproportionate amount of the enzyme to the lysosome rather than the secretory pathway. Secreted and lysosomal SMPD1 are likely to play distinct roles in the body[16], and abnormal levels of the secreted form have been implicated in age-related neurodegenerative conditions[17] including Alzheimer's disease[18] and amyotrophic lateral sclerosis (ALS)[19]. We, therefore, identify a locus at *GNPTAB* that coregulates secreted SMPD1 levels and *GNPTAB* expression, pinpointing a possible mechanism behind SMPD1-related neuropathological disorders.

**Heritability**. To estimate the narrow-sense heritability of the protein traits studied, the proportion of variance explained (PVE) by all variants across the genome was calculated using GCTA GREML[20] for each protein. Using a single-component approach, WGS variants explained a median of 33.3% of variance in serum protein levels, with the highest observed heritability observed for CD33 ($h^2$ = 87.2%). Another three proteins had high heritability of more than 80%: TDGF1 (85.4%), VSTM1 (82.8%), and LAIR2 (82.3%). Conversely, some proteins had very low heritability estimates of $h^2$ < 5%: IKZF2 (4.9%), RNF31 (4.4%), and EPHA10 (0.001%).

We observe that for all four proteins with $h^2$ > 80%, the pQTLs colocalised with gene expression QTLs in multiple tissues, indicating regulation at the transcriptional level; therefore, the high observed $h^2$ values are likely to mirror genuine high heritability. There are, however, other non-mutually exclusive reasons that can drive very high or low estimates: (1) Variants that alter the binding specificity of the Olink antibody but not the quantity of protein may produce inaccurate heritability estimates; and (2) Known and unknown biases of single-component GREML approach, which tends to overestimate $h^2$ when causal variants are common, and underestimate $h^2$ when causal variants are rare[21] (Supplementary Fig. 1).

**Link to disease outcomes**. To explore the biological relevance of the pQTLs, we carried out colocalisation analysis with neuropsychiatric traits using data published by the Psychiatric Genomics Consortium (PGC), as well as other neurodegenerative traits, using publicly available summary statistics from recent large GWAS meta-analyses (Supplementary Data 5b). We also studied colocalisation with signals for pain-related traits that have been proven to have a neuropathic component, such as chronic back pain[22] and osteoarthritis[23]. A total of 15 protein–trait pairs colocalised with human disease signals, suggesting a role for the protein in mediating disease. These results are summarised in Supplementary Data 5a.

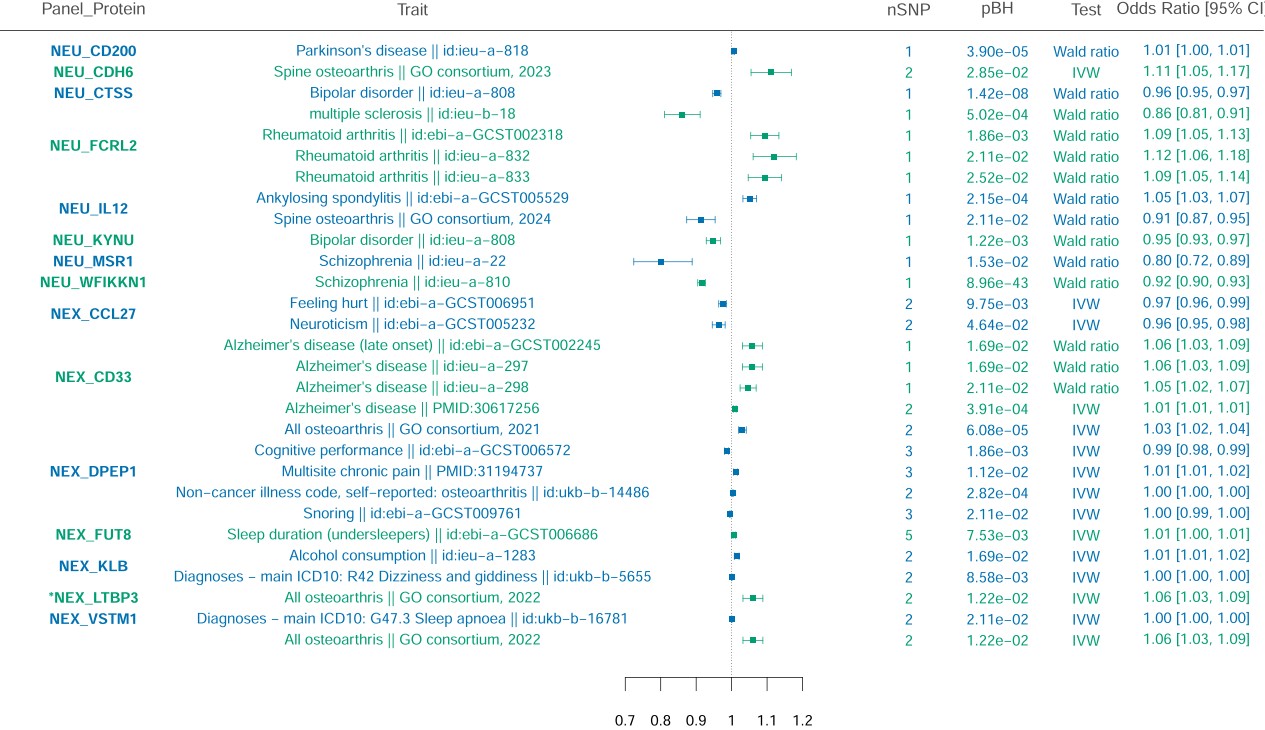

**Fig. 3 Causal protein-disease associations identified using two-sample Mendelian randomisation.** We investigated the causal effect of serum proteins (exposure) on various neurological traits (outcome), indicated in the first two columns in the plot. PubMed IDs (PMIDs) are given where manually downloaded summary statistics were used; other IDs are those as given in MRBase (https://gwas.mrcieu.ac.uk/). The number of variants used in the analysis are given in the 'nSNP' column. The 'pBH' column contains the FDR-adjusted (Benjamini–Hochberg) *P*-value for each test. Protein–trait pairs with only one variant were analysed using the Wald ratio method, while those with more than one variant were analysed using the inverse variance-weighted (IVW) method. Data are represented as mean odds ratio ± SEM. *Additional signal arising from analysis using only *cis*-pQTLs as instrumental variables.

We applied two-sample Mendelian randomisation (MR) for the 107 proteins for which we detect pQTLs, and 206 neurologically relevant and behavioural traits. In contrast to colocalisation, the objective of MR is to look for causal effects of proteins on neurological phenotypes. Using both *cis* and *trans*-acting pQTLs, fifteen proteins were found to be causal for at least one trait, and we detect significant causal effects for 25 unique protein–trait pairs (Fig. 3 and Supplementary Data 6a).

We replicate multiple known associations between protein and disease from the colocalisation and MR analyses. These include LEPR (leptin receptor) and migraine[24], LTBP3 (latent-transforming growth factor beta-binding protein 3) and osteoarthritis[25], FLRT2 (leucine-rich repeat transmembrane protein) with bipolar disorder[26], and PLXNB1[27] (plexin-B1) and PLA2G10[28] (group 10 secretory phospholipase A2) with schizophrenia.

The analysis also identified new protein-disease relationships. Notably, the strongest causal association was found between serum WFIKKN1 and schizophrenia ($P_{adj} = 9.12 \times 10^{-43}$); WFIKKN1 (WAP, Kazal, immunoglobulin, Kunitz and NTR domain-containing protein 1) has not been associated with any neuropsychiatric disorder to date, but is highly expressed in the brain (GTEx) and regulates the activity of several growth and differentiation factors[29]. Similarly, we find new evidence that serum VSTM1 is causally associated with sleep apnoea ($P_{adj} = 2.03 \times 10^{-2}$). VSTM1 (V-set and transmembrane domain-containing protein 1) is a cytokine that promotes the differentiation of helper T-cells (TH17), which are often implicated in autoimmune disorders that may develop secondary to sleep apnoea[30,31].

The overarching aim of this study was to identify protein biomarkers that may be used in the prognosis, diagnosis, or treatment of neurological diseases. Here, we highlight various potential disease markers that are supported by multiple lines of evidence.

**GPNMB as a biomarker for Parkinson's disease.** We identified a *cis*-pQTL that is associated with decreased levels of serum GPNMB (transmembrane glycoprotein NMB; rs7797870; MAF = 0.4286; $P = 7.01 \times 10^{-50}$; BETA = −0.2109; SE = 0.0247) and colocalises with a known Parkinson's disease (PD) locus[32] (CLPP4 = 0.86) (Fig. 4b). *GPNMB* has been highlighted as a susceptibility gene in large PD meta-analyses[32] and has been proven to be upregulated in the brains of PD patients and in mice with induced lysosomal dysfunction[33]. In addition to its connection to PD, we present new evidence showing that serum GPNMB shares a causal variant with *GPNMB* gene expression in both whole blood (CLPP4 = 0.79) and brain tissue (basal ganglia CLPP4 = 0.70; cortex CLPP4 = 0.74; anterior cingulate cortex CLPP4 = 0.83). This not only implies that GPNMB expression is regulated transcriptionally by the pQTL, but also that its expression in the blood and brain are mediated via a shared mechanism. This is supported by previous research showing that tissue GPNMB is able to shed its ectodomain and enter circulation[34]. The lead variant rs75801644 explained 7% of variance in antibody binding for serum GPNMB. Importantly, the identification of serum GPNMB levels as a potential marker of PD is significant as current diagnostic biomarkers are mostly found in the CSF. As serum biomarkers are much less invasive to

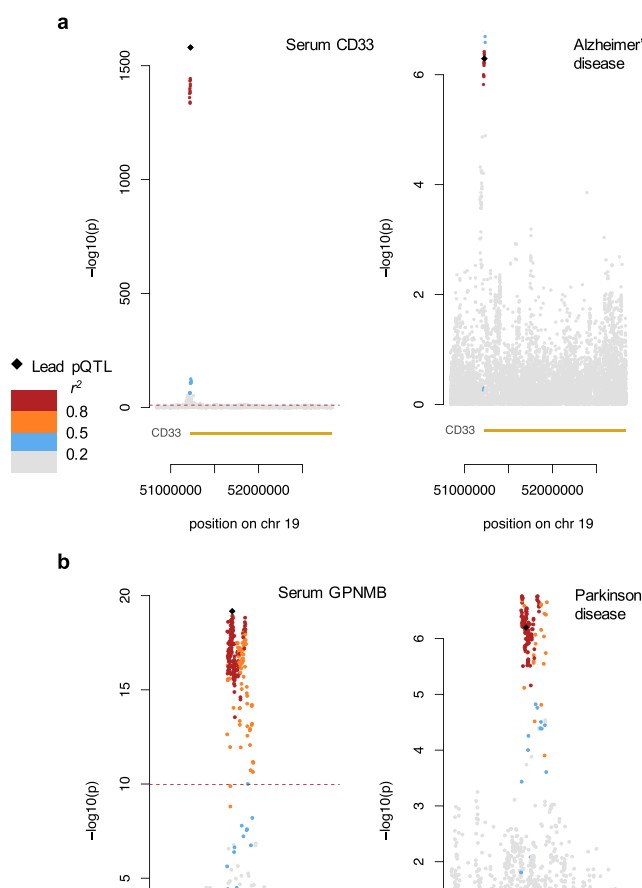

**Fig. 4 Colocalisation plots. Each plot shows the association signal and the −log10 P-values.** The lead pQTL variant is represented by a black diamond, while other points are variants that are coloured according to the extent of linkage disequilibrium with the lead variant. The location of the genes of interest are also shown in yellow at the bottom of each plot. Significance thresholds used for each respective study are shown using a dotted red line. **a** Left: Protein QTL signal for serum CD33; right: GWAS signal for Alzheimer's disease. **b** Left: Protein QTL signal for serum GPNMB; right: GWAS signal for Parkinson's disease.

measure, they are generally preferred for routine testing or monitoring disease progression. Clinical studies will be required to evaluate translational utility.

**CD33 as a biomarker for Alzheimer's disease**. Using two-sample MR, we confirm a significant causal association between serum CD33 (myeloid cell surface antigen CD33) and Alzheimer's disease[35] (AD; BETA = 0.0091; SE = 0.0017; inverse variance-weighted [IVW] $P_{adj}$ = 3.62 × 10$^{-4}$) (Figs. 3 and 4a). The role of CD33 in AD is further affirmed by positive colocalisation between the *cis*-pQTL with the causal variant rs2455069 (MAF = 0.3967; $P$ = 2.03 × 10$^{-1580}$; BETA = 1.2092; SE = 0.0142), and a known AD-associated locus (CLPP4 = 0.82). CD33 is upregulated in the AD brain and is positively correlated with disease severity, while knockout mice have been shown to have reduced amyloid plaque

formation[36]. Additionally, the *cis*-pQTL for CD33 colocalises with an eQTL for the *CD33* gene in whole blood (CLPP4 = 0.95) and in the brain (cerebellar hemisphere CLPP4 = 0.62), indicating a shared regulatory pathway for gene and protein expression.

Notably, our heritability analysis revealed a very high $h^2$ value (82.7%) for serum CD33, which is the highest proportion of variance explained observed across all analysed traits, thus reflecting high heritability. This has been similarly observed in a study showing that the most strongly AD-associated variant in *CD33*, rs3865444, explained more than 70% of variance in CD33 monocyte expression and was moreover unaffected by age[37]. A reverse Mendelian randomisation analysis (using AD as the exposure and serum CD33 as the outcome) confirmed that AD is causal for increased CD33. Together, these findings indicate that serum CD33 levels are a promising diagnostic marker for early AD (Supplementary Note 2).

**MSR1 on the causal pathway to schizophrenia**. We find that a *cis*-pQTL (rs150158578) associated with decreased serum MSR1 (macrophage scavenger receptor types I and II) is causal for schizophrenia, supported by evidence from colocalisation analysis (CLPP4 = 0.75) and two-sample MR (BETA = −0.2205; SE = 0.0522; Wald ratio $P_{adj}$ = 1.44 × 10$^{-2}$; Fig. 3). Variants in the MSR1-encoding gene have been nominally significantly associated in a schizophrenia GWAS[38], and have been robustly associated with AD[39] and PD[38].

MSR1 is an immune modulator expressed on the cell surface of macrophages. The protein plays a critical role in the clearance of infectious agents and toxic molecules, such as amyloid-beta protein[40], damage-associated molecular patterns (DAMPs)[41], and modified lipids, such as oxidised low-density lipoprotein (oxLDL)[42]. MSR1-mediated phagocytosis activates both pro- and anti-inflammatory responses, and has been shown to have a protective effect against multiple diseases, including bacterial and viral infections, AD, atherosclerosis and Barrett's oesophagus (BE)[43]. Accordingly, MSR1-deficient mice have been shown to exhibit dysregulated immune response in the brain and deteriorating working memory[44]. MSR1 activation can also lead to excessive inflammation linked to sepsis and worsening the cardiac and cerebral injury. Here, we observe a causal association between decreased MSR1 expression and increased risk of schizophrenia, suggesting a protective role (Fig. 5b).

We also find colocalisation of the *cis*-pQTL for serum MSR1 with an eQTL for the *MSR1* gene in the nucleus accumbens of the basal ganglia (CLPP4 = 0.90), aorta (CLPP4 = 0.94), tibial artery (CLPP4 = 0.90), and oesophagus (CLPP4 = 0.91) (Fig. 5c). The nucleus accumbens is central to the brain's reward system, and is enriched in dopaminergic neurons that contribute to the pathophysiology of schizophrenia[45,46] and other neuropsychiatric diseases[47,48]. A large comorbidity study has shown that patients with schizophrenia are more likely to suffer from coronary heart disease, cerebrovascular disease, and congestive heart failure[49]. We observed no evidence of colocalisation or causality between serum MSR1 and stroke or coronary artery disease (CAD).

To further investigate the mechanism through which the pQTL regulates protein expression, we queried the ENCODE[50] (https://www.encodeproject.org/) database for overlaps with *cis* regulatory elements. We found that, in blood cells, three variants in LD ($r^2$ > 0.8) with rs15015857 (rs420931, rs433235, and rs59251421) reside within regulatory elements with a proximal enhancer-like signature (EH38E2612565), a promoter-like signature (EH38E2612567), and a distal enhancer-like signature (EH38E2612573), respectively. All three variants, as well as rs150158578, are also eQTLs for *MSR1* gene expression in whole blood (GTEx). This suggests that the pQTL regulates MSR1 in

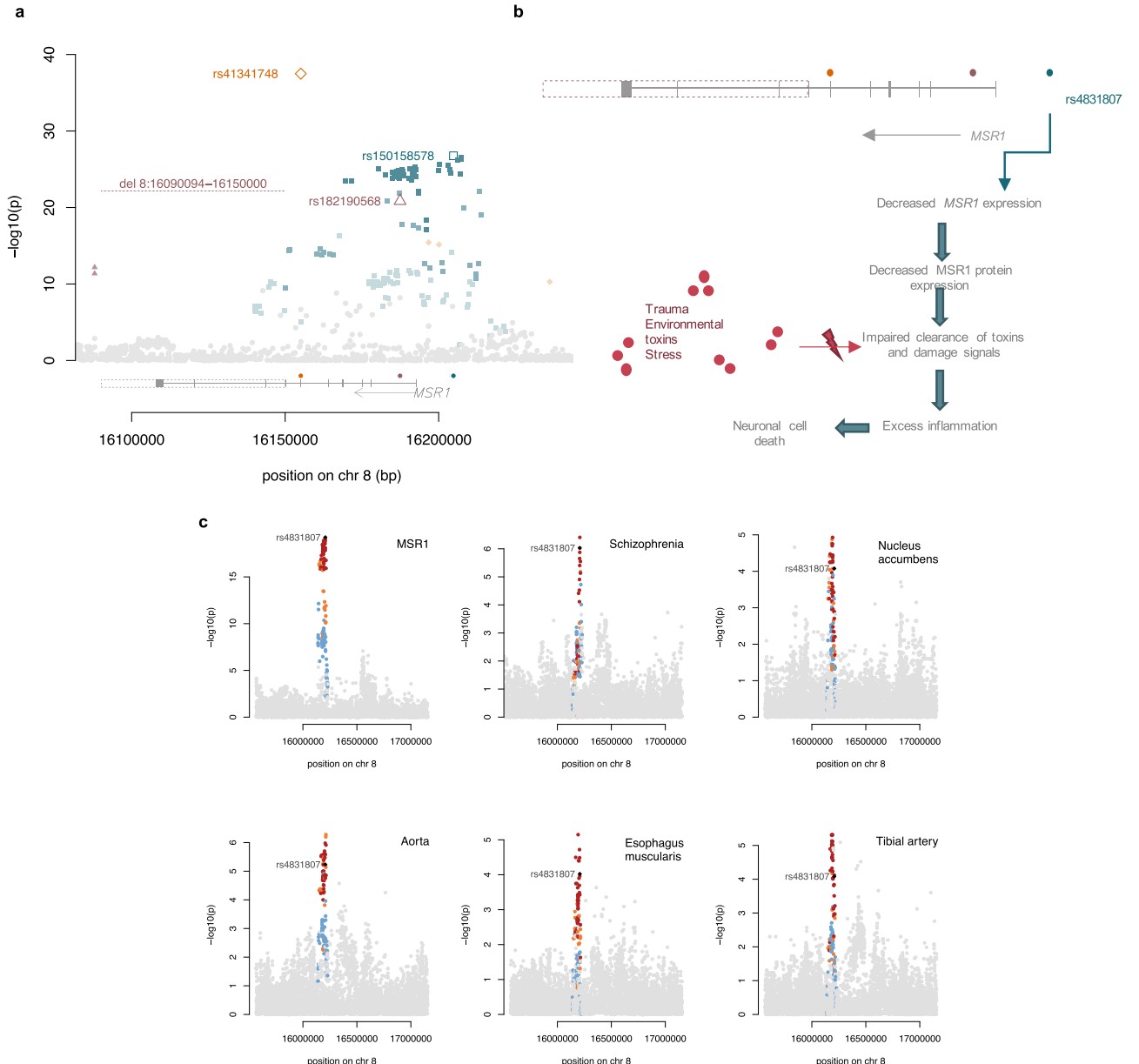

**Fig. 5 Serum MSR1 is causally associated with schizophrenia. a** Genetic architecture of serum MSR1. Each of the three independent variants and their LD variants are represented in orange, teal, and purple, respectively; the intensity of the colours indicates the strength of linkage disequilibrium ($r^2$). A rare deletion is also indicated in purple, and is in complete LD with the independent variant rs182190568. Below the signal plot, the location of the variants respective to the gene are indicated using coloured points for the SNVs and a dotted box for the deletion. **b** Proposed mechanism of how decreased MSR1 may lead to neuronal damage, resulting in neuropsychiatric disease. **c** Association signal plots at the MSR1 locus for (clockwise) serum MSR1, schizophrenia, gene expression of *MSR1* in nucleus accumbens tissue, aorta, oesophagus muscularis and tibial artery. The lead pQTL variant is denoted by a black diamond, while variants in LD are coloured according to the strength of LD with the lead variant (red [$r^2 > 0.8$]; orange [$0.5 < r^2 > 0.8$]; blue [$0.2 < r^2 > 0.5$]; grey [$r^2 < 0.2$]).

blood cells at the transcriptional level, possibly by altering the binding affinity of transcription factors to the promoter or an enhancer. Additionally, we note two other *cis*-acting rare, independent variants (rs182190568, MAF = 0.006, $P = 1.44 \times 10^{-21}$, BETA = −1.2568, SE = 0.1317; rs41341748, MAF = 0.0148, $P = 3.18 \times 10^{-38}$, BETA = −1.3351, SE = 0.1033), and a rare deletion (chr8: 16090094-16150000[b38]; MAF = 0.006; $P = 7.10 \times 10^{-23}$; BETA = −1.414, SE = 0.1436) that are significantly associated with serum MSR1 levels (Fig. 5a), illustrating the complexity of the genetic regulation of MSR1.

**Drug target evaluation**. Drug repositioning can dramatically expedite translational applications of proteomics and genomics into patient benefit. As over 95% of drugs target proteins[51], we sought to identify proteins included in this study that are targets of drugs that have been approved, or are in later stages of clinical trials (see 'Methods' section). Twenty-three of the proteins we studied in this work are targets of approved drugs. Of these, 17 proteins had pQTL signals (Supplementary Data 7).

Seven of these proteins have *cis*-acting pQTLs that colocalise with or are causal for neurological diseases: DDR1, IL12, NEP,

CD33, DPEP1, GPNMB, and LEPR (Supplementary Note 3). Of note is DPEP1 (dipeptidase 1), whose increased expression is causal for osteoarthritis and multisite chronic pain (MCP) (Fig. 3). DPEP1 is inhibited by the drug cilastatin, which is often used in combination with the antibiotic imipenem as an embolic agent in the treatment of serious infections. Given that DPEP1 is causally associated with osteoarthritis, cilastatin could potentially be repurposed to treat osteoarthritis. Indeed, the cilastatin/imipenem combination has been investigated as a treatment for knee osteoarthritis[52,53], and has been proven to provide pain relief. Also notable is CD33, whose expression is increased in AD (Figs. 3 and 4a). CD33 has proven to be a safe target, demonstrated by the acute myeloid leukaemia (AML) drugs, gemtuzumab ozogamicin and lintuzumab. In a study investigating the repurposing of lintuzumab for reducing AD risk, the anti-CD33 drug was shown to robustly decrease cell surface expression of the protein[54]. We, therefore, provide further genetic evidence supporting repositioning of lintuzumab for AD treatment.

## Discussion
Biomarker discovery is a process central to precision medicine, and is especially important for many neurological disorders that remain challenging to diagnose and treat. Serum proteins make ideal intermediate traits to study as they are druggable, measurable targets that are strongly linked to both causative genetic variants and medical outcomes. Having knowledge of their underlying genetic architecture and how that may correlate with diseases can also enhance our understanding of disease aetiology. We have carried out a pQTL analysis of 184 neurologically relevant serum proteins using WGS data. Altogether, we find 214 pQTLs for 107 proteins, of which 33 were for proteins that are being analysed for the first time. We detect novel pQTLs for previously studied proteins and replicate established associations of both blood and CSF pQTLs.

Through downstream analysis, we highlight disease-relevant, translatable pQTLs by presenting new evidence supporting protein-disease associations; most notably, CD33 and Alzheimer's disease, GPNMB and Parkinson's disease, and MSR1 and schizophrenia. Additionally, we observed that serum DPEP1 is causal for both osteoarthritis and multisite chronic pain (MCP). Pain is the main symptom of osteoarthritis, and osteoarthritis is the leading cause of pain and disability worldwide[55]. DPEP1 has been implicated in osteoarthritis through a large genome-wide association study[56], and was additionally shown to be downregulated in mouse models of osteoarthritis[57]. These findings indicate that serum DPEP1 may serve as a valuable candidate biomarker for identifying patients with undiagnosed osteoarthritis and suffer from MCP.

Consistent with previously published pQTL studies, the majority (73.8%) of variants were intergenic and intronic; we also observed 31 (14.4%) variants in regulatory regions and 25 (11.6%) coding variants, either missense or with more severe consequences. We note a limitation of epitope-based proteomic assays, in that cis-acting protein structure-altering variants may affect epitope-binding affinity and, in turn, measured protein levels. We identified 21 proteins with cis-acting variants, which were either highly correlated ($r^2 > 0.8$) with, or were missense or more severe consequence variants themselves. Of these, 16 variants were determined by Olink to be within a possible epitope-binding site (Supplementary Data 3), including those for CD33, GPNMB, and MSR1. Further errors may also be introduced due to cross-reactivity and unspecific binding[58]. For 20 of 21 proteins with corresponding protein quantification using the SomaScan technique (an aptamer-based proteomic technology binding to

varying protein sites), the correlation between Olink and SomaScan[59] plasma protein measurements was evaluated in 485 individuals from the Fenland cohort[60], using Spearman's rank-based correlation (Supplementary Data 3). Notably, we observed good correlation in protein abundance between the two measurements for CD33 ($\rho = 0.60$), GPNMB ($\rho = 0.51$), and MSR1 ($\rho = 0.74$). Explanations for a lack of correlation are manifold, including missing specificity of the aptamer or antibody for the selected target, the low affinity of the aptamer, targeting of different protein isoforms, a different dynamic range of the assays, as well as other technical factors as recently summarised[61]. Further orthogonal validation using epitope-independent assays is warranted.

We detect no pQTLs for 77 proteins and only trans-pQTLs for 16 proteins. This may be explained by other limitations including those related to epitope-binding. Firstly, only proteins in the serum were quantified. As serum contains multiple cell types originating from different tissues, pQTL detection is volatile to changes in serum composition. We note, for example, that the cis-pQTL for CD33 is also a known blood cell QTL (rs3865444)[37], highlighting how different cell-type composition and therefore, sample handling, can affect the serum proteome and drive pleiotropic signals. Secondly, the individuals included in this analysis are of European ancestry only, and variants that are absent or present in extremely low frequencies in our cohorts would not have been detected. Therefore, our findings—in both the pQTL discovery and downstream causal inference analyses—cannot be extrapolated to non-European populations. Finally, our sample size may not be adequate for the detection of rare variants of small effect sizes, again stressing the importance of larger, ethnically diverse studies.

In conclusion, we present the results of the first WGS-based pQTL analysis of neurologically relevant serum proteins to date. In addition to exploring the genetic architecture of these proteins, we show that pQTL analysis has the potential to identify disease-relevant serum biomarkers for debilitating neurological conditions. We identify opportunities for the repurposing of therapeutic targets, and deliver deeper insight into disease pathways. We recognise that an effective biomarker must be able to differentiate similarly presenting disorders to avoid misdiagnoses; hence, special attention must be given to further validation. Finally, we provide a resource that may be utilised by future studies to develop new hypotheses and advance our understanding of brain-related disorders.

## Methods
**Cohorts and samples**. The two cohorts included in this analysis, MANOLIS and Pomak, are part of the Hellenic Isolated Cohorts (HELIC; https://www.helmholtz-muenchen.de/itg/projects-and-cohorts/helic/index.html). The HELIC study focuses on the genetics of complex traits, making use of characteristics of founder populations, such as increased frequency of rare variants, extended linkage disequilibrium, and reduced haplotype complexity. For MANOLIS, biological samples were collected from the mountainous Mylopomatos villages in Crete, Greece; whereas, Pomak refers to a set of mountainous villages in the North of Greece. Further phenotypic and genetic characteristics have been described in detail in previous publications[62–64]. The study was approved by the Harokopio University Bioethics Committee, and informed consent was obtained from all human subjects.

**Sequencing and variant calling**. Both MANOLIS and Pomak followed the same sequencing, alignment, and variant calling pipeline. Genomic DNA (500 ng) from 1482 MANOLIS samples and 1642 Pomak samples were sheared to a median size of 500 bp and subjected to standard Illumina paired-end DNA library construction. Adapter-ligated libraries were amplified by six cycles of PCR and subjected to DNA sequencing using the HiSeqX platform (Illumina) according to the manufacturer's instructions. Basecall files for each lane were transformed into unmapped BAMs using Illumina2BAM, marking adapter contamination and decoding barcodes for removal into BAM tags. PhiX control reads were mapped using BWA Backtrack and were used to remove spatial artefacts. Reads were converted to FASTQ and aligned using BWA MEM 0.7.8 to the hg38 reference (GRCh38) with decoys (HS38DH). The alignment was then merged into the master sample BAM file using

Illumina2BAM MergeAlign. PCR and optical duplicates are marked using bio-bambam markduplicates and the files were archived in CRAM format. Per-lane CRAMs were retrieved and reads pooled on a per-sample basis across all lanes to produce library CRAMs; these were each divided into 200 chunks for parallelism. GVCFs were generated using HaplotypeCaller v.3.5 from the Genome Analysis Toolkit (GATK) for each chunk. All chunks were then merged at sample level, samples were then further combined in batches of 150 samples using GATK CombineGVCFs v.3.5. Variant calling was then performed on each batch using GATK GenotypeGVCFs v.3.5. The resulting variant callsets were then merged across all batches into a cohort-wide VCF file using bcftools concat.

**Proteomics and QC.** Proteins from Olink's (https://www.olink.com) Neurology and Neuro-exploratory panels were measured in the serum of 1457 MANOLIS and 1611 Pomak samples. The full list of 184 proteins is provided in Supplementary Data 8. Protein expression was quantified using Olink's Proximity Extension Assay (PEA) technology. Briefly, each protein assay uses pairs of oligonucleotide-labelled antibody probes; when these antibody pairs bind to the target antigen, the oligo-nucleotides hybridise due to their proximity and are extended by DNA polymerase. These DNA barcodes are amplified by PCR and quantified using microfluidic qPCR. Protein expression levels are reported as Normalised Protein Expression (NPX) values, Olink's relative quantification unit, which is in the Log2 scale. NPX values are derived by adjusting raw qPCR Ct values against several internal controls —an extension control, inter-plate control, and a correction factor calculated using a negative control. Additionally, the negative control determines the limit of detection (LOD) for each assay, calculated as the negative control plus three standard deviations. We included all proteins and all below-LOD NPX values in our analysis. Fifty-two and 37 MANOLIS samples, and 68 and 60 Pomak samples failed vendor QC for the Neurology and Neuro-exploratory panels, respectively, and were excluded from the analysis. Reported NPX values were then rank-based inverse normal transformed (INT) and used for the association analysis.

**Association analysis and meta-analysis.** A maximum of 1365 samples from MANOLIS and 1537 samples from Pomak were analysed for the Neurology panel; and for the Neuro-exploratory panel, a maximum of 1372 samples from MANOLIS and 1545 samples from Pomak were analysed. For each cohort, whole genome-wide association analysis with 184 proteins was performed using a linear mixed model implemented in GEMMA v.0.94[65], simultaneously adjusting for covariates —age, sex, season of sample collection, plate number, plate row, and plate column. An empirical relatedness matrix was also used for each cohort to account for population structure; this was calculated on an LD-pruned set of low-frequency and common variants (MAF > 1%) that passed the Hardy–Weinberg equilibrium test ($P > 1 \times 10^{-5}$). Following per-cohort analysis, 12,392,022 variants common to the two cohorts were meta-analysed using the fixed-effects inverse variance-based method in METAL[66]. As no proteins displayed significant genomic inflation (0.95 < λ < 1.03), no genomic control was applied.

**Conditional analysis to identify independent variants.** Using the PeakPlotter software (https://github.com/hmgu-itg/peakplotter), we detected 171 signals. We observed several signals extending over large regions that were mistakenly broken up into multiple signals; because of this, 12 signals were excluded to give 159 signals. Independent variants were identified using the approximate conditional and joint stepwise model selection, implemented using the -slct option in GCTA-COJO[67], using a collinearity cut-off of 0.9. Before that, however, variants were first subjected to clumping in Plink 1.9[68] (www.cog-genomics.org/plink/1.9/), using a $r^2$ threshold of 0.1 and a clumping window of 1 Mb; this reduces the number of variants input to COJO to avoid overfitting of the model. We arrived at a final number of 214 independent variants for 140 signals after filtering for minor allele count (MAC) > 10, Hardy–Weinberg equilibrium $P > 1 \times 10^{-5}$, and replication (meta-analysis $P$-value < per-cohort $P$-value) in both cohorts.

**Significance thresholds**
*Single variant-based association and rare variant analysis.* For single variant-based association, the significance threshold was adjusted for multiple testing by correcting for the effective number of protein traits ($M_{\mathrm{eff}}$) and variants ($N_{\mathrm{eff}}$) analysed. The effective number of proteins was computed using the ratio of the eigenvalue variance to its maximum[69,70]:

$$M_{\mathrm{eff}} = M(1 - (M-1)V_{\lambda_{\mathrm{obs}}}/M^2) = 1 + \frac{tr(\Sigma^T\Sigma)}{M} \qquad (1)$$

where $V_{\lambda_{\mathrm{obs}}}$ is the variance of the eigenvalues of the correlation matrix. For the $M = 184$ Olink proteins included in the study, $M_{\mathrm{eff}} = 93$ in both cohorts. The effective number of variants, or $N_{\mathrm{eff}}$, was determined by using the --indep and --maf options offered in Plink 1.9 to prune these variants. Specifically, variants with a minor allele count (MAC) of <10 were excluded; and parameters specified for --indep were: window size of 50 kb, variant count of 5, and variance inflation factor (VIF) of 2. This was performed separately for both the MANOLIS and Pomak cohorts, with resulting $N_{\mathrm{eff}}$s of 5,078,182 and 4,144,062 in each respective cohort. The more conservative $N_{\mathrm{eff}}$ of 5,078,182 was considered for the calculation of the $P$-

value significance threshold for the meta-analysis to give a final $P$-value threshold of $1.05 \times 10^{-10}$. The same threshold was used for the rare variant analysis.

*Significance threshold for two-sample MR.* $P$-values were adjusted for multiple testing by controlling for false discovery rate (FDR) using the Benjamini–Hochberg method. Results were considered significant if the FDR-adjusted $P$-values were below 0.05.

**Novelty.** We assessed variants for novelty using a funnel approach, by first identifying (a) novel proteins, then (b) novel signals, and finally, (c) novel variants. Novel proteins were defined as proteins that are being analysed for pQTLs for the first time. This was determined by comparing our proteins against protein lists from four large pQTL studies[7,10–12,71], querying GWAS Catalogue for known signals, then confirmed by doing manual literature searches. Next, we determined variants belonging to novel signals by checking against previously reported pQTLs (Supplementary Data 2). Signals were considered novel if no variants had been reported within 1 Mb upstream and downstream of our variants. All variants from known loci were then assessed for novelty by matching their rsIDs against previously reported variants; where no match was found, variants were conditioned on other known variants at the locus, and considered novel if the association $P$-value remained significant after conditioning.

**Heritability.** Heritability analysis was performed using GCTA GREML[20] (https://cnsgenomics.com/software/gcta/index.html#GREML), using both the multi-component LDMS and single-component approaches in two separate cohorts. The final meta-analysis $h^2_{\mathrm{meta}}$ was calculated using the following formula (provided on the GCTA website):

$$h^2_{\mathrm{meta}} = \sum(h_i^2/\mathrm{SE}_i^2)/\sum(1/\mathrm{SE}_i^2), \mathrm{SE} = \sqrt{(1/\sum(1/\mathrm{SE}_i^2))} \qquad (2)$$

**GREML-LDMS.** For each cohort, the segment-based LD score was first calculated using GCTA's --ld-score-region with the default length segment of 200 Kb. Variants were then stratified into four quartiles according to their LD scores in R, and a genetic relatedness matrix (GRM) was calculated for each group. For each protein, we then ran REML analysis with four GRMs using default settings. REML analysis failed to converge for 45 proteins across the two cohorts, likely due to limitations arising from a smaller sample size.

**GREML-SC.** As we were unable to obtain $h^2$ estimates for all proteins using GREML-LDMS, we also ran single-component GREML (GREML-SC) for all protein traits using a single GRM (also computed using GCTA). Full results may be found in Supplementary Data 9.

**Variant consequences.** We used Ensembl's variant effect predictor[72] (VEP; http://www.ensembl.org/vep) to determine the most severe consequence of each variant. To check for potential protein-altering effects, we also queried the most severe consequence of variants in LD ($r^2 > 0.8$) with reported *cis*-acting variants, which were extracted using PLINK 1.9. Variants with, or in LD with variants with potentially protein-altering consequences are reported in Supplementary Data 3.

**eQTL colocalisation.** Colocalisation analysis was performed using our pQTL results and gene expression QTL (eQTL) data downloaded from the GTEx database (https://www.gtexportal.org/), using the coloc.fast function from the gtx R package (https://github.com/tobyjohnson/gtx/). The method is equivalent to coloc by Giambartolomei et al.[73] and assumes only one causal variant at each associated locus. To satisfy this assumption in our pQTL data, for each independent variant, we conditioned associations on all other independent variants at the locus. For *cis*-pQTLs, we tested colocalisation with an expression of the encoding gene in all available tissues. For *trans*-pQTLs, colocalisation was performed with all genes within 2 Mb of the causal variant for all available tissues. For all analysed genes, eQTL data within 1 Mb upstream and downstream of the causal variant was extracted.

**PheWAS colocalisation.** Using the same conditioned pQTL data from the eQTL colocalisation analysis, we performed colocalisation with psychiatric and neuro-degenerative traits. For each analysed locus, GWAS data within 2 Mb of the causal variant was extracted. We used only publicly available summary statistics, either downloaded from the Psychiatric Genomics Consortium (PGC) website (https://www.med.unc.edu/pgc/download-results/), or as mentioned in the respective papers. A list of studies used can be found in Supplementary Data 5b. Additionally, colocalisation analysis was carried out with PhenoScanner[74,75] traits of neurological relevance. The results for this are included in Supplementary Data 5a. Five different posterior probabilities are reported in the table (CLPP0-CLPP4), which corresponds to the five tested hypotheses explained in Giambartolomei et al.[73]. In particular, CLPP4 indicates association with both tested traits with a shared causal variant.

**Two-sample Mendelian randomisation**. Two-sample MR was performed between 107 protein traits and 206 neurologically relevant phenotypes, using the TwoSampleMR R package[76] (https://github.com/MRCIEU/TwoSampleMR). Traits available in the MRBase[77] platform were selected based on the following: (a) Self-reported traits in UK Biobank with at least 1000 cases; (b) UK Biobank ICD10 primary and secondary traits of neurological relevance; (c) studies categorised as 'Psychiatric/neurological', 'Personality', and 'Sleeping'; (d) other large neurologically relevant traits with more than 10,000 samples; (e) manually downloaded summary statistics (see 'PheWAS colocalisation' section). Independent variants with an association meta-analysis $P < 5 \times 10^{-8}$ were determined by GCTA-COJO (see 'Peak calling and independent variants' section) and used as instrumental variables (IV), including both $cis$ and $trans$ variants. All variants at pleiotropic loci, including $KLKB1$, $FUT2$, $ABO$, $ST3GAL6$, and the HLA region, were excluded from the analysis. For each protein–trait pair, pQTL summary statistics for all independent variants and their variants in LD ($r^2 > 0.8$) were first extracted, excluding those without rsIDs. This was than harmonised with the available outcome data. Where any independent variant was not available in the outcome data, an LD variant ($r^2 > 0.8$) was used as proxy instead. For protein traits with more than 1 causal variant (IV), we used the inverse variance-weighted method; otherwise, Wald ratio estimates were used. Sensitivity analysis was carried out for protein–trait pairs with more than one IV by assessing heterogeneity about the IVW estimate using Cochran's $Q$ tests, with $P < 0.05$ denoting significant heterogeneity. We find that none of the protein–trait pairs with an FDR-adjusted $P < 0.05$ had Cochran's $Q$ $P < 0.05$. The analysis was also repeated using only $cis$-pQTLs (Supplementary Data 6b). This resulted in an additional causal signal, LTBP3 with osteoarthritis; and the loss of three signals: ADAM23 with neuroticism, NEP with osteoarthritis, and SIGLEC1 with osteoarthritis. We note an important caveat of our analysis, which is that when only one instrumental variable is available, a higher risk of violating the two-sample MR assumptions exists. Results from Wald ratio tests should, therefore, be interpreted cautiously and with orthogonal validation.

**Drug target evaluation**. Drug target evaluation was done by querying the Open Targets[78] (https://www.targetvalidation.org/) and Drugbank[79] (https://go.drugbank.com/) databases (Supplementary Data 7).

**Ethics statement**. The study was approved by the Institutional Review Board of Harokopio University and the Greek Ministry of Education, Lifelong Learning and Religious Affairs. The MAN-OLIS and Pomak studies were approved by the Harokopio University Bioethics Committee and informed consent was obtained from every participant.

**Reporting summary**. Further information on research design is available in the Nature Research Reporting Summary linked to this article.

## Data availability
The MANOLIS sequencing data used in this study are available at the European Genome-Phenome Archive (EGA) under accession number EGAS00001001207. The Pomak sequencing data have not been deposited to the EGA as the data and the information derived from it are culturally and politically sensitive in the context of this religiously isolated population. We will consider requests to access the data by researchers when an alternative cohort cannot reasonably be used for their research, and will respond to such requests within 6 months. Summary statistics generated in this study are available for download in the GWAS Catalogue. Accession codes and the respective hyperlinks are provided in Supplementary Data 10.

## Code availability
Analysis was performed using publicly available software as described in the 'Methods' section. Additional scripts may be found in our GitHub repositories (https://github.com/hmgu-itg).

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

## Acknowledgements

HELIC-MANOLIS study: We thank the residents of the Pomak and Mylopotamos villages for taking part. This work was funded by the Wellcome Trust [098051] and the European Research Council [ERC-2011-StG 280559- SEPI]. TEENAGE study: The TEENAGE study has been supported by the Wellcome Trust (098051), European Union (European Social Fund—ESF) and Greek national funds through the Education and Lifelong Learning Operational Program of the National Strategic Reference Framework (NSRF)—Research Funding Program: Heracleitus II, Investing in knowledge society through the European Social Fund. The GATK3 programme was made available through the generosity of the Medical and Population Genetics Program at the Broad Institute, Inc. We thank the Human Genetics DNA Pipelines and Human Genetics Informatics departments at the Wellcome Sanger Institute for performing sequencing and variant calling. This study has been conducted using the UK Biobank Resource (project ID 10205). J.F.W. and P.N. acknowledge support from the MRC Human Genetics Unit programme grant, 'Quantitative traits in health and disease' (U. MC_UU_00007/10). The Fenland Study (10.22025/2017.10.101.00001) and C.L., E.W. and N.J.W. are funded by the Medical Research Council (MC_UU_12015/1). We are grateful to all the volunteers and to the General Practitioners and practice staff for assistance with recruitment. We thank the Fenland Study Investigators, Fenland Study Co-ordination team and the Epidemiology Field, Data and Laboratory teams. We further acknowledge support for genomics and metabolomics from the Medical Research Council (MC_PC_13046). Proteomic measurements were supported and governed by a collaboration agreement between the University of Cambridge and Somalogic.

## Author contributions

Sample collection and phenotyping: E.T., M.K., G.D. and E.Z. Phenotype transformation and quality control: G.P., L.R., P.N., X.S. and J.F.W. Association analysis: G.P. Downstream bioinformatics: G.P., A.G. and A.B. Somalogic comparison with Fenland cohort: M.P., E.W., N.J.W. and C.L. Software development: A.G. and A.B. Study design and supervision: E.Z. Manuscript writing: G.P. and E.Z. Manuscript editing: G.P., A.B., L.R., P.N., X.S., M.P., E.W., N.J.W., C.L., E.T., M.K., G.D., A.M., J.F.W., A.G. and E.Z.

## Competing interests

The authors declare no competing interests.
