## [Peer Review File · Nature Communications]

Mapping the serum proteome to neurological diseases using whole genome sequencingReviewers' Comments:

Reviewer #1:

Remarks to the Author:

This is a comprehensive GWAS of neurological disease related proteins with extensive down-stream analyses of pathway and implications of these proteins with neurological disease. Paper is well composed with a number of novel findings and confirmation to existing studies. My comments are the follows:

1. The authors should provide a list of all proteins investigated in this study.
2. What is the total number of variants analyzed in each cohort? How effective number of variants N_{eff} was determined?
3. Using the proportion of variance explained by lead variants to estimate the heritability of the protein traits is not adequate. The heritability may also be contributed by variants that did not reach the significance threshold.
4. Why the number of proteins is 91 in mendelian randomization (MR) analyses?
5. Using only study-wide significant variants in the MR analyses may be too conservative. Many of the MR analyses were only using one variant. Lower the threshold to include slightly more associated variants may be more convincing.

Reviewer #2:

Remarks to the Author:

The increasing burden of neuropsychiatric traits on the health of an aging world population is a critical concern, thus there is an increasingly urgent need for new drug treatments and diagnostic gene markers. To this end Png et al. collected a large set of WGS from nearly three thousand individuals along with quantification for a limited set of serum proteins with the goal of identifying genes with causal links to neuropsychiatric phenotypes. With this putative causal list of genes, the authors further speculate on which may serve as likely candidate targets for putative drug therapies. Despite the constraints of the study (limited number of proteins targeted), the analysis seems mostly well done and the findings clearly written out. Still, I have a few major comments that should be addressed.

Major comments

I appreciate that the linear model for pQTL analysis included an empirical relatedness matrix to account for population structure. Other similar e/pQTL studies also include one or several principal components (PCs) which can further eliminate confounding from population structure and sometimes increase power. Was there a reason not to include PCs computed from genotypes? Does including some number of PCs increase or decrease the number or significance of identified pQTL associations? I assume such an analysis is particularly important with isolated populations such as these.

Protein traits with PVE higher than 25% only had cis signals, which the authors use as evidence that these genes could be less influenced by environmental factors. However, the method for estimating heritability apparently only relies on the top associated variant (was their no aggregation of heritability explained from multiple SNPs even if there were multiple independent signals?), which can underestimate contributions from many small effects unlikely reaching the significance threshold. If the authors would like to keep the statement about the influence of environmental factors I would suggest, if possible, adapting a GCTA-based approach for estimating the genetic contribution to protein heritability from all identified SNPs and not only the genome-wide significant hits.

The rationale behind the selection of the 184 proteins selected for quantification was unclear aside from being "neurologically-relevant". Why were these proteins selected? What evidence exists that support they are neurologically relevant? Is there evidence that genetics strongly contribute to their expression from eQTL evidence? It could be that this set of genes was a preselected panel, but I

would still appreciate the rationale behind choosing this panel over perhaps a custom set of genes. This is a particularly important point since it limits the scope of the study.

Minor comments

The introduction was lacking appropriate citations. For example, please cite the studies that have established that neuropsychiatric traits have a substantial genetic component (Line 45)? Which specific studies do the authors use to justify that protein expression levels exhibit a stronger genetic component than non-molecular traits (Line 57)? These were two examples of statements without citations, please go through and check to properly cite such statements in the rest of the text.

The current study was limited to isolated Greek populations. Most large-scale GWAS were performed only including individuals of European descent, could the authors address whether, or how well, these findings would port to individuals of different ancestry?

Reviewer #3:

Remarks to the Author:

This paper presents a genome-wide association study (GWAS) of 184 blood circulating proteins determined on the Olink affinity proteomics platform. This is not the first GWAS of this kind, but it adds substantially to the growing knowledgebase of protein QTLs, especially for a number of neurologically relevant proteins that have never been studied in a GWAS before. The study is conducted following state-of-the-art approaches. The manuscript is very clearly written, and the results are well presented. Several interesting and health relevant cases are discussed in good detail, and without too much speculation. These are of relevance for neurological medicine, including research into Alzheimer's disease. I only have minor comments.

Specific comments

Line 30: "Here, we show that the human serum proteome is an accessible reservoir of potential biomarkers" is certainly an already well established concept, as this is by far not the first study with serum proteomics. I suggest to be a bit more specific in summarizing the main findings of this study.
Line 31: I would not consider WGS at 15x as "deep" whole genome sequencing and suggest removing this qualifier.

Line 34: I suggest to remove "neurological" as an adjective to the proteins on the Olink panel. These are blood circulating proteins that have, among other things, some functions related to some neurological pathways. The naming of the panel by Olink was mainly done for marketing reasons. The term "related to neurological pathways/function" (as done later in the paper) is more adequate.

Introduction: Even if well established, some of the statements in the intro should be supported by references.

Line 53: Add "serum" to "The human [SERUM] proteome is an especially valuable resource of potential biomarkers ...". The proteome in general is much more than a resource for biomarkers.

Line 154-157: Comparing the explained variance of proteins with that of some random non-blood traits feels a bit like comparing apples with pears. Also, I don't think that the "hypothesis that proteomic traits have a stronger genetic component than non-molecular traits" needs further investigation. It is largely a matter of which traits are selected for comparison. The authors actually found no genetic component for 77 proteins, so it is not generally true anyway. I suggest removing this part.

Note that opening of Supplementary Table 1 in Excel lead to the following warning on my computer: "Security Warning: External Data Connections have been disabled [Enable Content]".

Figure 2: Increase the character size – the presentation is not very clear, as the legend dominates the picture – maybe better represented as a table?

Line 166: Up to this point nothing has been said about possible other explanations for the high explained variance of some of the proteins, i.e. the presence of epitope-changing variants that would not change the protein levels per se, but merely the antibody binding affinity. Also, cross-reactivity or

simple miss-annotation by OLINK of an antibody's target could explain some of the trans-only hits. These caveats should be introduced at an early stage, not be hidden in the supplement. For a discussion of these issues see for example PMID 32860016. This paper also contains the list of pGWAS referred to by a blog post in the Supplement.

Line 238: It is not clear why sleep apnoea is mentioned in this context. Does the protein signal also colocalize with a signal on this trait?

Figure 5: Increase the character size

Line 369: It is not clear how "moreover proving to provide pain relief" is related to the rest of the phrase – is this additional information from the investigation into the drug or is it a hypothesis?

Line 400: Why was one protein not compared to those in the Fenland study? Can you comment on the lack of correlation for some of the proteins (GSTP1, PSG1, IL32). Is difference in blood matrix (serum vs. plasma) a possible reason?

Line 593: I acknowledge that changing the significance threshold at this point is not reasonable. However, I personally feel that this kind p-value adjustment is kind of overkill, if it is to eventually change an initially arbitrary significance threshold by a factor of two. I would not do this in future studies and stick to simple Bonferroni correction.

Mapping the serum proteome to neurological diseases using whole genome sequencing

Thank you for the constructive comments and the opportunity to revise our manuscript. Please find our point-by-point response to comments below.

Reviewer #1 (Remarks to the Author):

This is a comprehensive GWAS of neurological disease related proteins with extensive downstream analyses of pathway and implications of these proteins with neurological disease. Paper is well composed with a number of novel findings and confirmation to existing studies. My comments are the follows:

1. The authors should provide a list of all proteins investigated in this study.

We agree with the reviewer and have now included the list of all proteins investigated in Supplementary Table 8.

2. What is the total number of variants analyzed in each cohort? How effective number of variants N_{eff} was determined?

The total number of variants analysed was 25,371,797 in MANOLIS, and 18,822,531 in Pomak. The effective number of variants, or N_{eff} , was determined by using the *--indep* and *--maf* options offered in Plink 1.9 to prune these variants. Specifically, variants with a minor allele count (MAC) of <10 were excluded; and parameters specified for *--indep* were: window size of 50kb, variant count of 5, and variance inflation factor (VIF) of 2. This was performed separately for both the MANOLIS and Pomak cohorts, with resulting N_{eff} s of 5,078,182 and 4,144,062 in each respective cohort. The more conservative N_{eff} of 5,078,182 was considered for the calculation of the P-value significance threshold for the meta-analysis. We have updated the Methods section (under 'Significance thresholds') with these details.

3. Using the proportion of variance explained by lead variants to estimate the heritability of the protein traits is not adequate. The heritability may also be contributed by variants that did not reach the significance threshold.

Thank you for the constructive feedback. We have now rerun the heritability analysis using GCTA GREML-SC, which considers variants across the whole genome. The full results may be found in Supplementary Table 9. Overall, heritability estimates using GREML were higher than our previous calculation of the proportion of variance explained. We observe some minor changes when ranking the proteins according to their heritability, although CD33 remains the protein with the highest heritability estimate. The Methods section has been updated with the following:

“Heritability analysis was performed using GCTA GREML⁶⁸, using both the multi-component LDMS and single-component approaches in two separate cohorts. The final meta-analysis h^2 was calculated using the following formula (as provided on the GCTA website):

$$h^2_{meta} = \text{sum}(h^2_i / SE^2_i) / \text{sum}(1 / SE^2_i) \text{ with } SE = \text{sqrt}(1 / \text{sum}(1 / SE^2_i))$$

GREML-LDMS:

For each cohort, the segment-based LD score was first calculated using GCTA’s `--ld-score-region` with the default length segment of 200Kb. Variants were then stratified into four quartiles according to their LD scores in R, and a genetic relatedness matrix (GRM) was calculated for each group. For each protein, we then ran REML analysis with four GRMs using default settings. REML analysis failed to converge for 45 proteins across the two cohorts, likely due to limitations arising from a smaller sample size.

GREML-SC

As we were unable to obtain h^2 estimates for all proteins using GREML-LDMS, we also ran single component GREML (GREML-SC) for all protein traits using a single GRM (also computed using GCTA). Full results may be found in Supplementary Table 9.”

4. Why the number of proteins is 91 in mendelian randomization (MR) analyses?

Only the 107 serum proteins for which we detect pQTLs were considered for the two-sample MR analysis. Of these, 16 proteins were further excluded as the only available instrumental variables were located in pleiotropic loci. Following the reviewer’s suggestion in point 5, we have rerun the MR analysis using a new set of instrumental variables. Both the main text and the methods section have been updated with the new results and a clearer explanation of these numbers.

5. Using only study-wide significant variants in the MR analyses may be too conservative. Many of the MR analyses were only using one variant. Lower the threshold to include slightly more associated variants may be more convincing.

We thank the reviewer for the suggestion. The two-sample MR analysis has been rerun using a lowered threshold of $P < 5 \times 10^{-8}$ to determine instrumental variables. The updated numbers may be found in the main text, and an updated MR table can be found in Supplementary Table 6.

The main text has been updated with the new numbers together with a new Figure 3:

“We applied two-sample Mendelian randomisation (MR) for the 107 proteins for which we detect pQTLs, and 206 neurologically relevant and behavioural traits. In contrast to colocalisation, the objective of MR is to look for causal effects of proteins on neurological phenotypes. Using both *cis* and *trans*-acting pQTLs, eighteen proteins were found to be causal for at least one trait, and we detect significant causal effects for 25 unique protein-trait pairs (Figure 3; Supplemental Table 6A).”

At the lowered threshold, each protein had 1-9 independent variants (cis-only), with each variant having up to 207 LD ($r^2 > 0.8$) proxies. We note that many of the comparisons still have only one available instrumental variable, despite us lowering the p-value threshold and using LD proxies where possible. In many cases, this happened when variants were not present in the GWAS data used for the two-sample MR. So as to not violate the MR assumption that instruments must be associated with the exposure, we did not relax the p-value threshold further than $P < 5 \times 10^{-8}$. Nevertheless, we agree that this is a caveat, and have included a note in the Methods section explaining the limitations of MR analyses using only one instrumental variable, as follows:

“We note an important caveat of our analysis, which is that when only one instrumental variable is available, a higher risk of violating the two-sample MR assumptions exists. Results from Wald ratio tests should, therefore, be interpreted cautiously and with orthogonal validation.”

Reviewer #2 (Remarks to the Author):

The increasing burden of neuropsychiatric traits on the health of an aging world population is a critical concern, thus there is an increasingly urgent need for new drug treatments and diagnostic gene markers. To this end Png et al. collected a large set of WGS from nearly three thousand individuals along with quantification for a limited set of serum proteins with the goal of identifying genes with causal links to neuropsychiatric phenotypes. With this putative causal list of genes, the authors further speculate on which may serve as likely candidate targets for putative drug therapies. Despite the constraints of the study (limited number of proteins targeted), the analysis seems mostly well done and the findings clearly written out. Still, I have a few major comments that should be addressed.

Major comments

I appreciate that the linear model for pQTL analysis included an empirical relatedness matrix to account for population structure. Other similar e/pQTL studies also include one or several principal components (PCs) which can further eliminate confounding from population structure and sometimes increase power. Was there a reason not to include PCs computed from genotypes? Does including some number of PCs increase or decrease the number or significance of identified pQTL associations? I assume such an analysis is particularly important with isolated populations such as these.

We did not include the principal components as covariates as PCs are typically computed by finding eigenvalues of the genetic relatedness matrix (GRM), which we used in our analysis. This means that the relatedness information contained in the full GRM is more complete than that in a small number of PCs. In linear mixed models such as GEMMA (which was used to run the association analysis), the matrix is used as is. This is different from approximate models such as SAIGE or REGENIE, where the matrix is thresholded. In such cases, the fine-grained relatedness structure would be lost, and it would then make sense to include PCs. In more homogeneous populations such as MANOLIS and Pomak ((PMID: 25373335), variation is spread across much finer family structures and we found that even up to 40 PCs were insufficient to capture this level of detail, as seen in the linear-like progression in the scree plot below.

PCA scree plot for 40 principal components in the MANOLIS cohort. The blue line represents the cumulative variance explained, while the red line represents the variance explained by each PC.

Nevertheless, to answer the question of whether including some PCs will change the number or significance of associations, we reran the association analysis for 92 proteins from the Neurology panel in the MANOLIS cohort using the first 20 PCs and the GRM. The analysis using 20PCs+GRM produced 73 peaks (using the Peakplotter software), while the GRM-only analysis produced 69 peaks. For the four peaks not detected in the GRM-only analysis, p-values for the most strongly associated variant fell just above the significance threshold (see table below). Comparing the p-values of 63 common lead variants, we found that p-values were significantly lower in the GRM-only analysis (Paired Wilcoxon signed rank test $P=5.983 \times 10^{-3}$).

Lead variants that were significantly associated ($P < 1.05 \times 10^{-10}$) in the 20PC+GRM analysis but not in GRM-only analysis

protein	chromosome	position	Pval.20PC+GRM	Pval.GRM_only
layn	11	111554933	6.56E-11	2.86E-10
scarf2	1	11493763	8.34E-11	1.67E-10
clm.1	17	77413121	9.83E-11	1.81E-10
alpha.2.mrap	6	39666486	5.19E-11	1.18E-10

Protein traits with PVE higher than 25% only had cis signals, which the authors use as evidence that these genes could be less influenced by environmental factors. However, the method for estimating heritability apparently only relies on the top associated variant (was their no aggregation of heritability explained from multiple SNPs even if there were multiple independent signals?), which can underestimate contributions from many small effects unlikely reaching the significance threshold. If the authors would like to keep the statement about the influence of environmental factors I would suggest, if possible, adapting a GCTA-based approach for estimating the genetic contribution to protein heritability from all identified SNPs and not only the genome-wide significant hits.

Thank you for the constructive advice. We have now rerun the heritability analysis using GCTA GREML-SC, which considers variants across the whole genome. The full results may be found in Supplementary Table 9. Overall, the heritability estimates using GREML are higher than our previous calculation of the proportion of variance explained (PVE), likely due to the aggregation of effects from multiple variants, as the reviewer has suggested. We observe some minor changes when ranking the proteins according to their heritability, although CD33 remains the protein with the highest heritability estimate. The Methods section has been updated with the following:

“Heritability analysis was performed using GCTA GREML⁶⁸, using both the multi-component LDMS and single-component approaches in two separate cohorts. The final meta-analysis h^2 was calculated using the following formula (as provided on the GCTA website):

$$h^2_{meta} = \frac{\sum(h^2_j / SE^2_j)}{\sum(1 / SE^2_j)} \text{ with } SE = \sqrt{1 / \sum(1 / SE^2_j)}$$

GREML-LDMS:

For each cohort, the segment-based LD score was first calculated using GCTA's `--ld-score-region` with the default length segment of 200Kb. Variants were then stratified into four quartiles according to their LD scores in R, and a genetic relatedness matrix (GRM) was calculated for each group. For each protein, we then ran REML analysis with four GRMs using default settings. REML analysis failed to converge for 45 proteins across the two cohorts, likely due to limitations arising from a smaller sample size.

GREML-SC

As we were unable to obtain h^2 estimates for all proteins using GREML-LDMS, we also ran single component GREML (GREML-SC) for all protein traits using a single GRM (also computed using GCTA). Full results may be found in Supplementary Table 9.

”

The rationale behind the selection of the 184 proteins selected for quantification was unclear aside from being “neurologically-relevant”. Why were these proteins selected? What evidence exists that support they are neurologically relevant? Is there evidence that genetics strongly contribute to their expression from eQTL evidence? It could be that this set of genes was a preselected panel, but I would still appreciate the rationale behind choosing this panel over perhaps a custom set of genes. This is a particularly important point since it limits the scope of the study.

The 184 proteins analysed were selected based on their relevance to neurological processes, and the information for each protein can be found on Olink's website (e.g., <https://www.olink.com/products/target/neurology/biomarkers/?biomarkerId=316>). The two panels comprise both established markers (such as CD33) and others with broader function, which widens the scope of discovery beyond known markers. Importantly, we opted for a technology that also measures the level of less abundant proteins.

Minor comments

The introduction was lacking appropriate citations. For example, please cite the studies that have established that neuropsychiatric traits have a substantial genetic component (Line 45)? Which specific studies do the authors use to justify that protein expression levels exhibit a stronger genetic component than non-molecular traits (Line 57)? These were two examples of statements without citations, please go through and check to properly cite such statements in the rest of the text.

Our apologies. The appropriate citations have now been added at the following lines:

- Multiple genetics and genomics efforts have established that these diseases have a substantial genetic component^{3,4}.
- Due to their heterogeneity and overlapping clinical features, neuropsychiatric disorders such as schizophrenia and bipolar disorder are often misdiagnosed⁵...
- ...while others with more distinct symptoms, such as Alzheimer's disease (AD), lack effective drugs and accessible biomarkers that can detect early disease⁶.

We have also deleted the sentence saying that proteins exhibit a stronger genetic component than non-molecular traits, upon realising that there are no published studies that have done in-depth comparisons on this thus far.

The current study was limited to isolated Greek populations. Most large-scale GWAS were performed only including individuals of European descent, could the authors address whether, or how well, these findings would port to individuals of different ancestry?

Thank you for the feedback. We have added a discussion of the limitations to the Discussion, as follows:

"We detect no pQTLs for 77 proteins and only trans-pQTLs for 16 proteins. This may be explained by other limitations including those related to epitope-binding. Firstly, only proteins in the serum were quantified. As serum contains multiple cell types originating from different tissues, pQTL detection is volatile to changes in serum composition. We note, for example, that the cis-pQTL for CD33 is also a known blood cell QTL (rs3865444)³⁷, highlighting how different cell-type composition and therefore, sample handling, can affect the serum proteome and drive pleiotropic signals. Secondly, the individuals included in this analysis are of European ancestry only, and

variants that are absent or present in extremely low frequencies in our cohorts would not have been detected. Therefore, our findings—in both the pQTL discovery and downstream causal inference analyses—cannot be extrapolated to non-European populations. Finally, our sample size may not be adequate for the detection of rare variants of small effect sizes, again stressing the importance of larger, ethnically diverse studies.”

Reviewer #3 (Remarks to the Author):

This paper presents a genome-wide association study (GWAS) of 184 blood circulating proteins determined on the Olink affinity proteomics platform. This is not the first GWAS of this kind, but it adds substantially to the growing knowledgebase of protein QTLs, especially for a number of neurologically relevant proteins that have never been studied in a GWAS before. The study is conducted following state-of-the-art approaches. The manuscript is very clearly written, and the results are well presented. Several interesting and health relevant cases are discussed in good detail, and without too much speculation. These are of relevance for neurological medicine, including research into Alzheimer's disease. I only have minor comments.

Specific comments

Line 30: "Here, we show that the human serum proteome is an accessible reservoir of potential biomarkers" is certainly an already well-established concept, as this is by far not the first study with serum proteomics. I suggest to be a bit more specific in summarizing the main findings of this study.

We thank the reviewer for the constructive feedback and agree that it is indeed an established concept. The particular sentence has been removed, and the abstract has been edited to be more specific. The updated abstract may be found in the response to the comment referring to Line 34.

Line 31: I would not consider WGS at 15x as "deep" whole genome sequencing and suggest removing this qualifier.

"Deep" has been removed from the sentence.

Line 34: I suggest to remove "neurological" as an adjective to the proteins on the Olink panel. These are blood circulating proteins that have, among other things, some functions related to some neurological pathways. The naming of the panel by Olink was mainly done for marketing reasons. The term "related to neurological pathways/function" (as done later in the paper) is more adequate.

We agree and have updated the sentence. The updated abstract is as follows:

"Despite the increasing global burden of neurological disorders, there is a lack of effective diagnostic and therapeutic biomarkers. Proteins are often dysregulated in disease and have a strong genetic component. Here, we carry out a protein quantitative trait locus (pQTL) analysis of 184 neurologically-relevant proteins, using whole genome sequencing (WGS) data from two isolated population-based cohorts (15x WGS; N=2,893). In doing so, we elucidate the genetic landscape of the circulating proteome and its connection to neurological disorders. We detect 214 independently-associated pQTLs for 107 proteins, the majority of which (76%) are cis-acting, including 114 pQTLs that have not been previously identified. Using two-sample Mendelian

randomisation, we identify causal associations between serum CD33 and Alzheimer's disease, GPNMB and Parkinson's disease, and MSR1 and schizophrenia, describing their clinical potential and highlighting drug repurposing opportunities.”

Introduction: Even if well established, some of the statements in the intro should be supported by references.

Thank you for pointing this out. We have added citations to the following sentences:

- Multiple genetics and genomics efforts have established that these diseases have a substantial genetic component^{3,4}.
- Due to their heterogeneity and overlapping clinical features, neuropsychiatric disorders such as schizophrenia and bipolar disorder are often misdiagnosed⁵...
- ...while others with more distinct symptoms, such as Alzheimer's disease (AD), lack effective drugs and accessible biomarkers that can detect early disease⁶.

Line 53: Add “serum” to “The human [SERUM] proteome is an especially valuable resource of potential biomarkers ...” The proteome in general is much more than a resource for biomarkers.

This has been added.

Line 154-157: Comparing the explained variance of proteins with that of some random non-blood traits feels a bit like comparing apples with pears. Also, I don't think that the “hypothesis that proteomic traits have a stronger genetic component than non-molecular traits” needs further investigation. It is largely a matter of which traits are selected for comparison. The authors actually found no genetic component for 77 proteins, so it is not generally true anyway. I suggest removing this part.

We agree with the reviewer and have removed the section comparing the heritability of proteins with blood traits.

Note that opening of Supplementary Table 1 in Excel lead to the following warning on my computer: “Security Warning: External Data Connections have been disabled [Enable Content]”.

Thank you for pointing this out and apologies for the oversight. This has now been fixed.

Figure 2: Increase the character size – the presentation is not very clear, as the legend dominates the picture – maybe better represented as a table?

We have increased the character size in Figure 2.

Line 166: Up to this point nothing has been said about possible other explanations for the high explained variance of some of the proteins, i.e., the presence of epitope-changing variants that

would not change the protein levels per se, but merely the antibody binding affinity. Also, cross-reactivity or simple miss-annotation by OLINK of an antibody's target could explain some of the trans-only hits. These caveats should be introduced at an early stage, not be hidden in the supplement. For a discussion of these issues see for example PMID 32860016. This paper also contains the list of pGWAS referred to by a blog post in the Supplement.

Thank you for the feedback. We agree and have added the following paragraph to the section on heritability discussing possible reasons driving very high or low heritability estimates:

“We observe that for the four proteins with $h^2 > 80\%$, the pQTLs colocalised with gene expression QTLs in multiple tissues, indicating regulation at the transcriptional level; therefore, the high observed h^2 values are likely to mirror genuine high heritability. There are, however, other non-mutually exclusive reasons that can drive very high or low estimates: (1) Variants that alter the binding specificity of the Olink antibody but not the quantity of protein may produce false signals and hence, inaccurate heritability estimates; and (2) Known and unknown biases of the method used (single-component GREML), which tends to overestimate h^2 when causal variants are common, and underestimate h^2 when causal variants are rare¹¹ (Supplementary Figure 3).”

An additional paragraph on study limitations has also been added to the discussion:

“We detect no pQTLs for 77 proteins and only *trans*-pQTLs for 16 proteins. This may be explained by other limitations including those related to epitope-binding. Firstly, only proteins in the serum were quantified. As serum contains multiple cell types originating from different tissues, pQTL detection is volatile to changes in serum composition. We note, for example, that the *cis*-pQTL for CD33 is also a known blood cell QTL (rs3865444)³⁷, highlighting how different cell-type composition and therefore, sample handling, can affect the serum proteome and drive pleiotropic signals. Secondly, the individuals included in this analysis are of European ancestry only, and variants that are absent or present in extremely low frequencies in our cohorts would not have been detected. Therefore, our findings—in both the pQTL discovery and downstream causal inference analyses—cannot be extrapolated to non-European populations. Finally, our sample size may not be adequate for the detection of rare variants of small effect sizes, again stressing the importance of larger, ethnically diverse studies.”

Line 238: It is not clear why sleep apnoea is mentioned in this context. Does the protein signal also colocalize with a signal on this trait?

Our apologies for the lack of clarity. In this context, our MR results had suggested a causal relationship between serum VSTM1 and our trait of interest, sleep apnoea. As this relationship had not been studied before, our aim was to offer a possible explanation for VSTM1's role in sleep apnoea. We found that both VSTM1 and sleep apnoea have been previously implicated in

Rheumatoid arthritis on separate occasions, and thus wanted to draw a link between VSTM1, sleep apnoea, and Rheumatoid arthritis. We have edited the paragraph as follows:

“Similarly, we find new evidence that serum VSTM1 is causally associated with sleep apnoea. VSTM1 is a cytokine that promotes the differentiation of helper T-cells (TH17), which are often implicated in autoimmune disorders that may develop secondary to sleep apnoea^{26,27}.”

Figure 5: Increase the character size

We have increased the character size in Figure 5.

Line 369: It is not clear how “moreover proving to provide pain relief” is related to the rest of the phrase – is this additional information from the investigation into the drug or is it a hypothesis?

We apologise for the lack of clarity. This is additional information from the referenced papers investigating the repurposing of the drug (cilastatin/imipenem) for treatment of osteoarthritis. The purpose of this sentence was to emphasise the optimistic results of current cilastatin/imipenem repurposing efforts. We have edited the paragraph as follows:

“Of note is DPEP1, whose increased expression is causal for osteoarthritis and multisite chronic pain (MCP) (Figure 3). DPEP1 is inhibited by the drug cilastatin, which is often used in combination with the antibiotic imipenem as an embolic agent in the treatment of serious infections. Given that DPEP1 is causally associated with osteoarthritis, cilastatin could potentially be repurposed to treat osteoarthritis. Indeed, the cilastatin/imipenem combination has been investigated as a treatment for knee osteoarthritis^{49,50}, and has been proven to provide pain relief”

Line 400: Why was one protein not compared to those in the Fenland study? Can you comment on the lack of correlation for some of the proteins (GSTP1, PSG1, IL32). Is difference in blood matrix (serum vs. plasma) a possible reason?

The protein that was not compared has no SomaScan data, hence no comparison could be made. The text has been edited for clarity. A full list of proteins measured by both technologies can also be found in Supplementary Table S1 of a recent paper by Pietzner et al. (bioRxiv DOI: 10.1101/2021.03.18.435919).

Regarding the second question, correlation was analysed using only data from the Fenland cohort, which uses only plasma samples. While we observed a good correlation of highlighted protein candidates with the complementary SomaScan technique, such as MSR1 (74%), explanations for a lack of correlation are manifold, including missing specificity of the aptamer or antibody for the selected target, low affinity of the aptamer, targeting of different protein isoforms, a different dynamic range of the assays, as well as other technical factors as recently summarised by Pietzner et al. (bioRxiv DOI: 10.1101/2021.03.18.435919). Nevertheless, the fact

that we were able to detect cis-pQTLs colocalising with gene expression QTLs for 18 of these proteins (see updated Supplementary Table S3) also provides validation for the target specificity of the Olink's technology.

The paragraph has been updated as follows:

“For 20 of 21 proteins with corresponding protein quantification using the SomaScan technique (an aptamer-based proteomic technology binding to varying protein sites), correlation between Olink and SomaScan⁵⁹ plasma protein measurements was evaluated in 485 individuals from the Fenland cohort⁶⁰, using Spearman's rank-based correlation (Supplemental Table 3). Notably, we observed good correlation in protein abundance between the two measurements for CD33 ($\rho=0.60$), GPNMB ($\rho=0.51$), and MSR1 ($\rho=0.74$). Explanations for a lack of correlation are manifold, including missing specificity of the aptamer or antibody for the selected target, low affinity of the aptamer, targeting of different protein isoforms, a different dynamic range of the assays, as well as other technical factors as recently summarised⁶¹. Further orthogonal validation using epitope-independent assays is warranted.”

Line 593: I acknowledge that changing the significance threshold at this point is not reasonable. However, I personally feel that this kind p-value adjustment is kind of overkill, if it is to eventually change an initially arbitrary significance threshold by a factor of two. I would not do this in future studies and stick to simple Bonferroni correction.

Thank you for the input. We have not changed the significance threshold used here, but can understand your perspective. Given that many variants are correlated, as are some proteins, our aim was to use a threshold that would be most appropriately adjusted for our data. We acknowledge, however, that the resulting adjusted P-value threshold is not hugely different from a Bonferroni-corrected P-value threshold (1.05×10^{-10} vs. 1.07×10^{-11}). We will use a Bonferroni-corrected P-value in future studies.

Reviewers' Comments:

Reviewer #2:

Remarks to the Author:

My concerns have been addressed.

Mapping the serum proteome to neurological diseases using whole genome sequencing

REVIEWERS' COMMENTS

Reviewer #2 (Remarks to the Author):

My concerns have been addressed.

Thank you!